# Protein Prenyltransferases and Their Inhibitors: Structural and Functional Characterization

**DOI:** 10.3390/ijms23105424

**Published:** 2022-05-12

**Authors:** Aleksandra Marchwicka, Daria Kamińska, Mohsen Monirialamdari, Katarzyna M. Błażewska, Edyta Gendaszewska-Darmach

**Affiliations:** 1Institute of Molecular and Industrial Biotechnology, Faculty of Biotechnology and Food Sciences, Lodz University of Technology, 90-537 Lodz, Poland; marchwicka.aleksandra@outlook.com (A.M.); daria.kaminska@dokt.p.lodz.pl (D.K.); 2Institute of Organic Chemistry, Faculty of Chemistry, Lodz University of Technology, 90-924 Lodz, Poland; mohsen.monirialamdari@dokt.p.lodz.pl (M.M.); katarzyna.blazewska@p.lodz.pl (K.M.B.)

**Keywords:** prenyltransferases, structures, inhibitors

## Abstract

Protein prenylation is a post-translational modification controlling the localization, activity, and protein–protein interactions of small GTPases, including the Ras superfamily. This covalent attachment of either a farnesyl (15 carbon) or a geranylgeranyl (20 carbon) isoprenoid group is catalyzed by four prenyltransferases, namely farnesyltransferase (FTase), geranylgeranyltransferase type I (GGTase-I), Rab geranylgeranyltransferase (GGTase-II), and recently discovered geranylgeranyltransferase type III (GGTase-III). Blocking small GTPase activity, namely inhibiting prenyltransferases, has been proposed as a potential disease treatment method. Inhibitors of prenyltransferase have resulted in substantial therapeutic benefits in various diseases, such as cancer, neurological disorders, and viral and parasitic infections. In this review, we overview the structure of FTase, GGTase-I, GGTase-II, and GGTase-III and summarize the current status of research on their inhibitors.

## 1. Introduction

Many regulatory proteins, which can be found on the membrane surface or incorporated into the lipid bilayer, require anchoring to cellular membranes to operate biologically. In most cases, post-translational modifications (PTMs) with lipid moieties convert nascent hydrophilic proteins to hydrophobic peripheral membrane proteins with higher affinity for the lipid bilayer. PTMs have received a lot of attention in recent decades because of their importance in a diversity of cellular processes and the regulation of protein activities. Protein prenylation, based on the chemical attachment of a farnesyl or geranylgeranyl anchor to conserved cysteine residues in the C-terminus of proteins, is unique among PTMs because of its structural plasticity and capacity to govern the attachment of soluble proteins to cell membranes [1]. 

The first prenylated protein was discovered in 1978 in *Rhodosporidium toruloides*, an oleaginous, carotenogenic basidiomycete yeast [2]. In turn, the first prenylated mammalian protein, lamin B, was identified in 1988 [3]. A cysteine residue in lamin B was shown to contain a thioether-linked farnesyl near the carboxyl terminus of the protein [4]. This discovery gave rise to a pioneering area of research focusing on the novel post-translational lipid modification of proteins. Since then, the number of recognized prenylated proteins has been continually growing thanks to novel analytical methodologies based on mass spectrometry and is predicted to affect hundreds of proteins in the human proteome [5]. Most prenylated proteins include the Ras-related G proteins (small GTPases), particularly Ras, Rab, and Rho. However, subunits of heterotrimeric G proteins, nuclear structural proteins such as lamins, glycogen metabolism enzymes, centromeric proteins, and cell cycle and apoptosis regulators also require prenylation to function [6]. Upon termination of translation, the C-terminal sequences of the proteins are recognized by a prenyltransferase. Four heterodimeric prenyltransferases are involved in this catalytic reaction, farnesyltransferase (FTase), geranylgeranyltransferase type I (GGTase-I), Rab geranylgeranyl transferase (GGTase-II, RGGT, RabGGTase), and recently discovered geranylgeranyltransferase type III (GGTase-III). These enzymes have been found in mammals, fungi, plants, and protists, among others [7,8,9].

In addition to its roles in normal cellular physiology, the protein prenylation process has crucial implications for human disorders. Many prenylated proteins have been linked to tumor formation and metastasis. The interest in prenylation increased when it was discovered that oncogenic Ras proteins undergo the above modification to obtain full functionality [10]. Aberrant protein prenylation is also implicated in cardiovascular, bone, and neurodegenerative disorders, including Alzheimer’s disease, and diabetes [1,11]. Therefore, the prenylation process has become an attractive target for inhibitors with therapeutic promise. Unfortunately, most proteins from the Ras superfamily are considered “undruggable” and require indirect targeting strategies. Therefore, targeting of small GTPases through the inhibitors affecting membrane localization by the inhibition of prenyltransferases has entered the spotlight [12]. Here, we discuss the mechanisms of small GTPase prenylation, give a structural and functional characterization of protein prenyltransferase, and present the status of prenyltransferase inhibitor research.

## 2. Overview of Protein Prenyltransferases and Their Substrates 

FTase, GGTase-I, and GGTase-II were discovered in the early 1990s [13,14,15,16]. FTase and GGTase-I, also called the CAAX prenyltransferases, transfer a single isoprenoid chain from farnesyl pyrophosphate (FPP) or geranylgeranyl pyrophosphate (GGPP), respectively, to the cysteine residue of the C-terminal CAAX motif of substrate proteins (Table 1), where A represents an aliphatic amino acid, while X stands for any amino acid residue and determines which isoprenoid chain will be attached. If it is alanine, methionine, serine, or glutamine, farnesylation occurs, while in the case of leucine, isoleucine, or phenylalanine, geranylgeranylation takes place [17]. An exception to this rule is the RhoB protein, which exists in both forms. Its C-terminal CKVL motif undergoes both farnesylation (30% of all RhoB molecules) and geranylgeranylation (70% RhoB). In the farnesylated form, the RhoB protein is responsible for cell growth, while in the geranylgeranylated form, it induces apoptosis [18]. FPP and GGPP are produced in the mevalonate pathway by farnesyl diphosphate synthase (FPPS) and geranylgeranyl diphosphate synthase (GGPPS), respectively [17].

Recent work supports the ability of yeast and mammalian FTase to modify sequences longer than the canonical CAAX sequence. Both can accept longer five-amino-acid C(X)3X sequences as substrates [19]. Using MALDI-MS analysis of peptide libraries, Distefano et al. showed that the A1 position revealed the greatest number of hits in the extended CAAAX libraries, possibly because it is the furthest away from the C-terminal X residue, which plays a significant role in substrate recognition. Many hits were also found in the A2 and A3 positions, with the A3 position displaying more conventional hydrophobic residues (Ala, Leu, and Met). With one exception (Lys), the X position, which is regarded as the most significant for peptide recognition, showed classical hits (Ala, Cys, Ser, Gln, and Met) [20]. Furthermore, FTase can also modify shorter CXX sequences and mammalian FTase can utilize GGPP as the prenyl donor for the modification of a subset of CXX sequences with efficiency comparable to that of the farnesylation of the identical sequences [21]. These data suggest that the prenylated proteome may be larger than previously thought.

Singly prenylated CAAX proteins undergo additional steps to increase their hydrophobicity and facilitate stable membrane attachment (Figure 1). After attachment of isoprenoid residues, the AAX tripeptide is cleaved by Ras converting enzyme 1 (RCE1) or Ste24. Then, the exposed carboxyl group is methylated with S-adenosylmethionine (SAM) by isoprenylcysteine methyltransferase (ICMT). Mature forms of S-prenylated proteins are then targeted and anchored to the appropriate membranes. Recently, it was shown that some prenylated proteins undergo a separate “shunt pathway” in which post-prenylation processing is bypassed and even demonstrated to be deleterious to the protein’s function [22]. 

Notably, recent efforts have been also made to target the postprenylation enzymes Rce1 and ICMT. Rce1 inhibition has raised some toxicity concerns and the selective inhibition of Rce1 remains a difficult task since the inhibition of functionally related Ste24 results in progeria, muscular dystrophy, and lipodystrophy [23]. ICMT itself is also a promising therapeutic target for many cancers since it is critical for malignant transformation in Ras-driven cancers [24]. 

Because the inactivation of ICMT prevents Ras from proper localization and activation, small-molecule inhibitors of ICMT have been developed and evaluated in preclinical and clinical trials for cancer treatment [25,26]. On the other hand, in some cases, a single prenylation may not be sufficient to immobilize proteins in membranes stably. Therefore, they may have a polybasic domain (PBR) or also be S-palmitoylated, e.g., p21Ras (Figure 1) [18,27]. Ras, RhoB, and Rheb GTPases, nuclear lamins, several protein kinases and phosphatases, and a few centromere proteins are farnesylated by FTase. FTase is also known to farnesylate the Golgi SNARE protein Ykt6 [28]. Subunits of trimeric G proteins, members of the Rho, Ral, and Rap family of small GTPases, and various other regulatory proteins are geranylgeranylated by GGTase-I. Interestingly, inhibition of FTase was connected to compensatory GGTase-I overexpression in the case of K-Ras, which could be a reason for anticancer FTase inhibitors’ lack of clinical success. As a result, dual FTase/GGTase-I inhibitors could be a more effective treatment option [29].

Rab geranylgeranyltransferase catalyzes the attachment of geranylgeranyl moieties to two neighboring cysteine residues located in the C-terminal sequences of Rab proteins, e.g., CC, CXC, CCX, CCXX, CCXXX (Table 1). Additionally, some Rab proteins containing the CXXX or CAAX motif may be monogeranylgeranylated by this enzyme [17]. In contrast to the CAAX prenyltransferases, RGGT requires the presence of Rab escort protein 1 or 2 (REP-1 or REP-2) to carry out prenylation. Through the C-terminal binding region (CBR), REP recognizes hydrophobic CBR-interacting motifs (CIM) in protein substrates, which then enables the formation of a ternary complex with RGGT. The REP protein, following the transfer of geranylgeranyl residues to Rab, facilitates the transfer of prenylated proteins to their target membranes, where they are anchored and transformed into the active form [18,30]. 

GGTase-III, the fourth form of protein prenyltransferase, has recently been discovered. The discovery of GGTase-III originates from the Human Genome Project, which revealed a novel protein with prenyltransferase homology, namely prenyltransferase α subunit repeat-containing 1 (PTAR1). PTAR1 is a prenyltransferase subunit and it joins with the RabGGTase β subunit (RabGGTB) to form the novel GGTase-III. GGTase-III catalyzes the double prenylation of the ubiquitin ligase FBXL2 and the Golgi SNARE protein Ykt6 along with FTase [9]. In addition, GGTase-III requires the chaperone SKP1 protein for geranylgeranylation [8]. Doubly prenylated Ykt6, but not singly prenylated Ykt6, is critical for efficiently sorting and trafficking acid hydrolases to lysosomes [31].

## 3. Structures of CAAX Prenyltransferases: FTase and GGTase-I and Mechanism of Prenylation

Prenyltransferases are heterodimeric, cytosolic enzymes containing α and β subunits. The first crystalline structure of the mammalian prenyltransferase was resolved in 1997, giving rise to structure–activity relationship (SAR) research to develop potent inhibitors of these enzymes. The crystal structure of the purified rat protein, showing 97% sequence identity to the human enzyme, was determined with a resolution of 2.25 Å (PDB: 1FT1) [32]. The structure of rat GGTase-I in complex with GGPP was determined in 2003 with a resolution of 2.65 Å (PBD: 1N4P) [33]. Both prenyltransferases show structural similarities and analogies in the catalyzed post-translational modification.

FTase and GGTase-I are heterodimers consisting of α and β subunits with sizes of 44 kDa and 49 kDa, and 44 kDa and 42 kDa, respectively (Figure 2). The amino acid sequences of the α subunits (FNTA encoded by the *FNTA* gene in humans) of both prenyltransferases are identical, while the β subunits (FNTB in FTase and PGGT1B in GGTase-I) share only 25% sequence identity. However, the homology between the central regions of FTase and GGTase-I is higher. The 3D structure of the α subunit is nearly identical for both prenyltransferases (Figure 2C). They differ only in the interaction site with the corresponding β subunits (the β subunit of GGTase I is smaller than FTase, containing 377 and 437 amino acid residues, respectively). The β subunits, on the other hand, although characterized by 25% sequence identity, have very similar secondary structures, consisting of fourteen α helices in FTase and thirteen in GGTase-I [7].

The secondary structure of prenyltransferases consists primarily of α helices (Figure 2). The helices 2–15 of the α subunit are arranged into seven pairs forming a series of right-handed, antiparallel coiled coil motifs. These helical structures are in turn organized into a crescent-shaped right-handed superhelix, which gives the α subunit the shape of a crescent surrounding a portion of the β subunit. All α helices in one layer are approximately parallel to each other and antiparallel to α helices in the adjacent layer. The N-terminal proline-rich domain of the α subunit, consisting of approximately fifty amino acids, does not form any secondary structures and is most likely responsible for the interaction with other cellular components, determining the location of the enzyme. Deletion of this sequence does not affect the catalytic activity or structure of the rest of the enzyme [32].

Twelve α helices of the β subunit form the α–α barrel motif. The core of the barrel is composed of six parallel α helices (3β, 5β, 7β, 9β, 11β, and 13β) connected by the remaining six α helices (2β, 4β, 6β, 8β, 10β, and 12β) forming the outer part of the barrel motif. The six outer α helices are parallel to each other and antiparallel to the six α core helices. One end of the barrel structure is blocked by a loop formed by the C-terminal amino acid residues of the β subunit (located at positions 399β to 402β in FTase). The opposite end is instead the solvent-accessible one. A deep hydrophobic pocket inside the α–α barrel, with an inner diameter of 15 Å and a depth of 14 Å, contains the active sites of CAAX prenyltransferases. The heterodimer is stabilized by extensive subunit interactions with the unusually high polar/charged interface that covers approximately 3300 Å^2^ (~20%) of the solvent-accessible surface area [7,32].

A single Zn^2+^ ion bound to the β subunit is present near the interface of the subunits and is coordinated by three conserved amino acid residues, Asp, Cys, and His (Figure 3). In FTase, Asp297β and Cys299β located in the N-terminal region of the loop surrounding the 11β helix, His362β in the 13β helix, and a water molecule are involved in Zn^2+^ coordination [32,34,35]. Asp359β was also shown to be involved in the binding of Zn^2+^ [35]. In the case of GGTase-I, Asp269β, Cys271β, and His321β bind Zn^2+^ [33]. The zinc ion is necessary to maintain proteins’ enzymatic activity and coordinate the cysteine thiol of the CAAX substrate or the thioether group in the product. In substrate complexes, the CAAX cysteine thiol(ate/ether) forms a short (2.3 Å) connection with Zn^2+^ and a longer (2.6 Å) interaction in the product complex [7]. Zinc is also not required for FPP binding, as revealed in the structure of the rat FTase lacking this ion in complex with the FPP analog and the peptide derived from K-Ras4B (CAAX motif—CVIM, PDB: 1D8E) [36]. Unlike GGTase-I, Mg^2+^ ions are key factors for the enzymatic activity of FTase. The binding of Mg^2+^ ions at the millimolar level increases the reaction rate 700-fold, possibly reducing the gap between the reacting partners [37,38]. Mg^2+^ interacts with phosphate groups of FPP and Asp352β. In GGTase-I, the Lys311β residue serves as the equivalent of Asp352β in FTase and its positively charged side chain enables stabilization of the diphosphate group [39].

Based on the high-resolution crystal structure of FTase with an FPP substrate (PDB: 1FT2) [40] and GGTase-I with GGPP (PDB: 1N4P) bound at the active site [33], it was shown that the binding sites for isoprenoid substrates are located in the cavity of the β subunit α–α barrel. This pocket has an inside diameter of 15 to 16 Å and a depth of 14 Å. In the case of FTase, the cavity contains ten highly conserved aromatic residues with which the isoprenoid interacts: Trp (102β, 106β, 303β), Phe (253β, 302β), and Tyr (105β, 154β, 205β, 361β, 365β). The farnesyl residue of FPP binds in an extended conformation interacting with conserved aromatic residues. Arg202β in the α helix 7β is positioned in such a way that allows it to interact with the phosphate residue of the substrate. After the FPP molecule is introduced into the pocket, the diphosphate group can interact with the zinc atom. In the GGTase-I active site, the first three isoprene units of GGPP bind similarly, while the fourth isoprene residue is inverted ~90° to the rest of the molecule. Introducing a GGPP molecule into FTase is not possible due to the presence of large amino acid residues such as Trp102β and Tyr365β in the depth of the binding pocket. In GGTase-I, their counterparts are the smaller Thr49β and Phe324β residues facilitating the binding of the fourth isoprene unit (Figure 4). The positively charged cleft is adjacent to the catalytic zinc ion and runs parallel to the α–α barrel near the subunit interaction surface and is limited by the twist connecting the 4α and 5α helices and the loops connecting the 8β–9β, 10β–11β, and 12β–13β helices. The diphosphate moiety of the FPP and GGPP molecules forms hydrogen bonds in a positively charged cleft with Lys164α, His248β, Arg291β, and Tyr300β in the case of FTase and Lys164α, His219β, Arg263, and Tyr2726ββ, Lys266β in GGTase-I.

FTase and GGTase-I recognize the same CAAX sequence motif in a protein substrate bound in the hydrophobic pockets of β subunits. The second and third isoprene units of the FPP analog and the fourth one in the GGPP analog are in direct contact with the last two amino acids of the CAAX motif and participate in van der Waals interactions. The modified cysteine residue directly coordinates the zinc ion. The co-crystallization of FTase and GGTase-I with FPP or GGPP analogs and peptides derived from K-Ras4B for FTase (CAAX motif—CVIM, PDB: 1D8D and 1QBQ) [36,41] or Rap2B for GGTase-I (CAAX theme—CVIL, PDB: 1N4Q) [33] revealed that the side chain of valine, located at the second position of the CVIM sequence of the peptide, does not interact hydrophobically with isoprenoid, in contrast to the side chain of the isoleucine residue (third position), which is in close proximity. The carbonyl oxygen of this residue also participates in hydrogen bonding with the enzyme (Arg202β in FTase, Arg173β in GGTase-I). Hydrogen bonds are also formed between the C-terminal carboxyl group of the X residue (Met in FTase, Leu in GGTase-I) and Gln167α and a water molecule coordinated by three amino acids of the enzymes (His149β, Glu198β, and Arg202β in FTase; His121β, Glu169β, and Arg173β in GGTase-I). The X residue also participates in van der Waals interactions with many other amino acids of the enzyme (Tyr131α, Ala98β, Ser99β, Trp102β, His149β, Ala151β, and Pro152β in FTase; Thr49β, His121β, Ala123β, and Phe174β in GGTase-I) [7,33,36].

The catalytic mechanism of the prenylation is very similar for FTase and GGTase-I. The binding of isoprenoid diphosphate occurs first. After forming the two-component complex, the protein substrate is aligned parallel to the FPP or GGPP molecule at the active site of a specific enzyme. The sulfhydryl group of the C-terminal cysteine residue of the CAAX motif reacts with the C1 atom of an isoprenoid residue to form a covalent thioether bond. The product’s release requires the binding of another substrate molecule—FPP or GGPP—which causes the prenyl group of the already modified protein to move to a new binding site, the so-called exit groove. The prenylated protein is then released from the complex before or during the binding of a new protein substrate [17,39].

## 4. Structure of Rab Geranylgeranyltransferase and Protein Complexes Involved in the Process of Rab Protein Prenylation and the Mechanism of Rab Prenylation

Rab geranylgeranyltransferase, as with other prenyltransferases, consists of two subunits: α with a molecular weight of 65 kDa and β with a mass of 37 kDa (Figure 5A). The sequence of the α subunit is 27% identical and the β subunit is 29% similar to the sequence of the corresponding FTase and GGTase-I subunits [17]. The RGGT crystal structure was first solved in 2000 with a resolution of 2 Å based on the rat enzyme expressed in Sf9 cells (PDB: 1DCE) [42].

Three domains can be distinguished in the structure of the α subunit of RGGT: the α-helical domain, the immunoglobulin-like (Ig-like) domain, and the LRR (leucine-rich repeat) domain containing leucine-rich repeats. The helical domain contains fifteen α helices (1α to 15α) forming a crescent-shaped right-handed superhelix. The presence of the other two domains in the α subunit is a feature that distinguishes RGGT from other prenyltransferases. They are not involved in the enzyme’s catalytic activity, and their function has not been understood so far. The Ig-like domain comprises amino acid residues 244α to 345α that form the structure of the eight-stranded β sandwich. It is located between the 11α and 12α helices and connected to the helical domain via two loops. One side of the domain (strands *h*, *a*, *b*, *e*) is positioned in proximity to the LRR domain. The amino acids at positions 443α to 567α, including the C-terminus of the α subunit, constitute the LRR domain. Its sequence includes five leucine-rich repeats ranging in length from 22 to 27 amino acid residues. The 3D structure of the LRR domain is formed by a right-handed superhelix consisting of β strands and 3_10_ helices (α helix in the last repeat) [42]. The more precise structure of the RGGT was revealed as a result of the crystallization of the enzyme without LRR and Ig-like domains (1.8 Å resolution, PDB: 3DSS). Slight shifts of the α helices of the α subunit were identified compared to the structure obtained for the native protein (Figure 5E). The conformation of the β subunit remained unchanged [43].

The β subunit of RGGT reveals a structure of an α–α barrel composed of twelve α helices, which is very similar to the analogous element found in FTase. Compared to the β subunit of FTase (437 amino acids), the smaller β subunit of RGGT (331 amino acids) lacks the first α helix and the long C-terminal loop. The center of the α–α barrel forms a funnel-shaped pocket, the interior of which is mainly filled with aromatic and hydrophobic amino acid residues. The bottom of the motif is blocked by a β turn showing a short α helix near the C-terminus of the β subunit. A positively charged area formed by Arg232β, Lys235β, and Lys105α is observed in the vicinity of the pocket. The substitution of FTase residues Trp102β and Tyr154β by Ser48β and Leu99β residues in RGGT is the major difference between the isoprenoid binding cavities in FTase and RGGT, forming a much wider and deeper binding pocket in the RGGT. Twenty-four amino acids surrounding the GGPP molecule in the GGTase-I binding pocket are identical to the corresponding residues in RGGT or are formed by structurally conserved equivalents. All the features mentioned above suggest a similar mode of isoprenoid binding in RGGT compared to other prenyltransferases, particularly GGTase-I. The helical domain of the RGGT α subunit surrounds the β subunit halfway around its circumference, near the open side of the α–α barrel pocket, while the Ig- domains are similar and LRR have no contact with it. The bottom of the α–α barrel and the 3_10_ helix side of the LRR domain form a distinct groove [42].

The presence of an intrinsic zinc ion in RGGT was identified, as in other prenyltransferases. It is located on the β subunit and is coordinated by Asp238β, Cys240β, and His290β. The fourth Zn^2+^ ligand is the His2α residue through which the N-terminal region of the α subunit interacts with the β subunit. Additionally, Lys6α interacts via ionic bonding with Asp272β. The N-terminal sequence of the α subunit also interacts with residues 283β–285β, which are part of the long loop connecting the 12β and 13β helices. The connection between the N-terminal end of the RGGT α subunit and its active site may be auto-inhibitory. It may prevent short peptide substrates from binding to the enzyme molecule. Nevertheless, the conformation of the 8α-26α amino acids is not rigid and allows the binding of the protein substrate [42].

The structure of the RGGT complex with GGPP (PDB: 3DST) revealed the binding mode of the lipid substrate (Figure 5F). The isoprenoid part is placed inside the α–α barrel filled with the aromatic amino acids, Tyr51β, Trp52β, Phe147β, Tyr195β, Tyr241β, Trp243β, Trp244β, Phe289β, Phe293β, Phe143α, and Tyr107α. The diphosphate residue is bound in a positively charged area formed at the site of subunit interaction, close to the zinc ion. The β-phosphate group participates in the formation of hydrogen bonds with Lys235β and Lys105α residues, while the α-phosphate group with Arg232β and a water molecule. Complex formation of GGPP with RGGT induces several minor changes in the structure of the active site of the enzyme, most of which involve rotation or displacement of the side chains of the hydrophobic Tyr195β, Cys240β, Tyr241β, Trp244β, and Phe293β residues. Compared to the GGTase-I complex with GGPP, the carbon atoms 1–7 of the RGGT-bound isoprenoid adopt a different conformation and are located closer to Zn^2+^. The position of the fragment of the molecule containing the 8–15 carbon atoms is identical, but the most significant difference is in the location of the last part of the GGPP. The lipid substrate binding site region surrounding the phosphate groups and the first 12 carbon atoms of GGPP comprise the same amino acids with similar positions in both enzymes. The binding pocket is less conserved in the depth of the α–α barrel pattern. Some substitutions do not lead to significant changes in the shape of the binding site or its hydrophobicity, e.g., Phe53β in GGTase-I on Trp52β in RGGT or Phe52β in GGTase-I on Tyr51β in RGGT. However, some differences significantly modify the properties of this pocket, e.g., the equivalents of the amino acids Leu320β, Tyr323β, Tyr126β, and Asn345β in GGTase-I are Phe289β, Leu292β, Leu99β, and Cys314β in RGGT, respectively. These changes are responsible for the curvature of the last isoprene unit of GGPP in the complex with the RGGT. Compared to GGTase-I, the lipid binding site in RGGT is enlarged in the depth of the cavity, mainly due to the substitution of tyrosine residues at positions 323β and 126β in GGTase-I. This makes RGGT more tolerant than the other prenyltransferases in binding isoprenoid analogs, e.g., biotin-labeled geranyl diphosphate used to identify prenylated proteins and to search for new inhibitors of this enzyme [43]. 

Thus far, it has not been possible to obtain a complete complex crystal consisting of all the elements involved in the geranylgeranylation of Rab proteins, i.e., RGGT, GGPP, and REP associated with a protein substrate or product (prenylated Rab protein). The interactions between individual components are known based on the structures of partial complexes, which also allowed the determination of the mechanism of Rab protein modification. The complex of the rat isoprenoid-bound RGGT and REP-1 with farnesyl phosphonyl(methyl)phosphonate (FPCP) (PDB: 1LTX) was solved to 2.7 Å resolution (Figure 6A). FPCP, a phosphatase-resistant analog of FPP, was used due to the failure of initial attempts to determine the structure of this complex in the presence of GGPP or FPP [44]. The complex interface between RGGT and REP-1 buries a small surface area of ca. 680 Å and is formed by the 8α, 10α, and 12α helices of the α subunit of the enzyme and helices D and E of the II domain of REP-1. The Arg290 of REP-1 participates in the formation of hydrogen bonds with the main chain of Phe220α and Thr221α RGGT, while Lys294 of REP-1 with Glu378αα of the 12α helix of the enzyme. The Met286 and Met291 of REP-1 are involved in the hydrophobic interaction with the Phe220α and Leu377α side chains of RGGT, respectively. Moreover, the side chain of Phe279 REP-1 targets deep into the cavity formed by the α helix 8α and 10α of the α subunit of RGGT. It is involved in the hydrophobic interaction with the Ile171α, Leu214α, Ala218α, and possibly Thr172α residues of the enzyme. Phe279 of REP-1 is the key amino acid responsible for forming the RGGT: REP-1 complex. The LRR and Ig-like RGGT domains do not interact with the REP-1 protein. The structure of RGGT in the complex with REP-1 is similar to that specified for the crystal of the enzyme itself. The only slight differences are conformational changes in the N-terminus of the α subunit and shifts in the positions of the α helices interacting with REP-1. In the case of the β subunit, conformational changes were observed at the aromatic amino acid residues Tyr241β, Trp244β, and His190β, which are located at the RGGT lipid binding site [44].

The presence of a lipid substrate also dramatically influences the formation of the RGGT:REP-1 complex. The binding of phosphoisoprenoid to the enzyme’s active site induces a series of small rearrangements that increase the affinity of REP-1 for RGGT. Most likely, after attaching a GGPP molecule to RGGT, an interaction is created between Arg144β and Tyr107α, which in turn causes a shift in the main chain of the 4α helix. It disrupts the contact between the 4α and 5α helices and causes a shift in the 6α and 8α helices. The new conformation of the 8α helix makes it possible to generate an interaction between REP-1 and RGGT via the Phe279 residue of the Rab escort protein [44].

Another crystal complex that facilitated the understanding of the mechanism of the RGGT-catalyzed reaction was formed by the rat REP-1 and monoprenylated Rab7 (PDB: 1VG0) (Figure 6C) or Rab7 C-terminally truncated by 22 amino acids (PDB: 1VG9) [45]. The crystal structures were solved to 2.2 Å and 2.5 Å resolution, respectively. Two binding interfaces of REP and Rab protein were identified. The globular part of Rab7 interacts with the Rab-binding platform (RBP) located on the REP-1 domain I, while its C-terminus is coordinated by the C-terminal-binding region (CBR) of REP. CBR consists of apolar amino acid residues forming a hydrophobic cavity upstream of the mobile effector loop (MEL). In the structure of the REP-1 complex with the truncated Rab7 protein, only the interaction between GTPase and RBP occurs, which indicates that the interaction with CBR is not necessary for complex formation, but its absence significantly reduces the affinity of the proteins. However, this region is required for the prenylation reaction. The strong interaction of the C-terminus of Rab proteins with CBR influences the association of the terminal cysteine with the enzyme’s active site. The CBR interacting motif (CIM), in which the presence of at least one hydrophobic amino acid is required for the binding of Rab to REP-1, is responsible for the interaction with CBR in Rab GTPases. The lack of strict rules regarding the amino acid composition of the CIM motif also explains the possibility of prenylation of various protein substrates by RGGT, which are characterized by high sequence variability at the C-terminus. In the REP-1:RGGT complex, the hydrophobic CBR cavity is occupied by the C-terminus of REP-1 in a manner analogous to the binding of the C-terminus of Rab7 in the complex with REP-1. 

The 1074 Å^2^ area of interaction between RBP and Rab7 consists of hydrophobic and electrostatic interactions. The stabilization of the complex is most influenced by Arg79, which is conserved in proteins from the Rab family and participates in the formation of hydrogen bonds with Asn225 and Glu379 of REP-1. In addition to Glu379, Arg386 of RBP was demonstrated to be critical for the function of REP-1. Furthermore, these amino acids are involved in forming hydrogen bonds with the conserved Asp44 and Asp63 residues of the Switch I and Switch II regions of Rab7 [45].

Further studies unraveled the specific role of REP in controlling the RGGT-catalyzed prenylation reaction. Guo et al. solved the structure of RGGT with a mono- or digeranylgeranylated Ser-Cys-Ser-Cys tetrapeptide corresponding to the C-terminus of the Rab7: Ser-Cys-Ser-Cys(GG), Ser-Cys(GG)-Ser-Cys, Ser-Cys(GG)-Ser-Cys(GG). These structures were used to identify the location of the conjugated isoprenoid during the second prenyl transfer reaction. In all three cases, the lipid-binding site was occupied by a geranylgeranyl moiety. At the same time, the electron density of the peptide chain was very low, indicating that there is no highly stabilizing interaction between the protein C-terminus and the enzyme active site. This explains the ability of RGGT to modify a wide variety of substrates in which cysteine residues do not need to be located at precisely defined positions in the C-terminus of Rab proteins. It may also be the reason for the requirement to recognize protein substrates by REP protein because the interaction between their C-terminus and the RGGT active site is too weak and would not arise spontaneously without additional factors. The introduction of a second prenyl residue into the Ser-Cys-Ser-Cys sequence reduced peptide flexibility [43]. 

Formation of the catalytic ternary Rab:REP:RGGT complex (Figure 7) begins with the recognition of the Rab GTPase domain by the RBP region of REP. The low-to-intermediate-affinity complex is further enhanced by the interaction of the CIM region with the CBR of REP. Thus, bound Rab is made available to RGGT. The enzyme has a bound GGPP molecule in its active site at this stage. The affinity of the complex is further enhanced by the weak and largely nonspecific interaction of the C-terminus of Rab with the RGGT active site. Finally, binding the C-terminal cysteine residue to the zinc ion in the RGGT starts the reaction [43].

Although RGGT can modify protein substrates with various C-terminal sequences, the lack of strictly defined interactions between proteins and their low affinity can negatively affect the rate of catalysis. RGGT is the slowest prenyltransferase (FTase catalyzed reaction rate expressed as *k*_chem_ value: 12-17 s^−1^, GGTase-I: 0.5 s^−1^, RGGT: stage I—0.16 ^s−1^, stage II—0.04 s^−1^). Upon attachment of the first isoprenoid group, the new GGPP molecule binds to the active site, causing a displacement of the monoprenylated intermediate. Despite the increase in the affinity of the elements of the complex after modification of the Rab protein with the first prenyl residue, the rate of the second step of the reaction decreases four-fold compared to the first step. The attached lipid residue may sterically hinder the reaction of modifying the second cysteine residue and explain the effect that occurs. Following the geranylgeranylation of the second thiol group, the doubly prenylated peptide is released. The modified C-terminus is displaced from the RGGT active site, inducing a conformational change in REP-1 domain II and thereby causing the RGGT complex to break down. The complex of REP and diprenylated Rab protein is then relocated to the target membrane in which the Rab protein will be anchored [43].

## 5. Structure of Geranylgeranyltransferase Type III

The discovery of a fourth prenyltransferase, namely geranylgeranyltransferase type III (GGTase-III), originates from the Genome Project, where prenyltransferase α subunit repeat containing 1 (PTAR1) was identified [8,9]. PTAR1 exhibits high homology to the Ftase and GGTase I (FNTA) and RGGT (RabGGTA) α subunits and is highly conserved from a fruit fly to mammals. It was observed that the F-box/LRR-repeat protein 2 (FBXL2), a component of Skp-Cullin-F box (SCF) ubiquitin E3 ligase, was co-immunoprecipitated in complex with the PTAR1 subunits, RabGGTB, or both, and separately with PTAR1 from RabGGTB. Using recombinant GGTase-III, the possibility of prenylating the FBXL2 protein, but not the K-Ras4B, was demonstrated. GGTase-I was also able to modify FBXL2 in vitro. Despite the presence of a typical C-terminal CAAX prenylation motif, FBXL2 is specifically recognized by GGTase-III in vivo. However, there is a possibility of GGTase-I geranylgeranylation of the F-box protein in the absence of GGTase-III activity. GGTase-III specifically geranylgeranylates FBXL2 and controls its cellular localization at cell membranes dependent on the C-terminal sequence of CVIL [9].

The GGTase-III:FBXL2: Skp1 complex was crystallized and its structure was determined to a resolution of 2.5 Å (PDB: 6O60) (Figure 8B). For the newly discovered GGTase-III, a strong high-affinity interaction was identified between the enzyme and FBXL2: Skp1, not requiring the presence of GGPP or an analog thereof. In the case of Ftase and GGTase-I, protein substrates bind to enzymes with high affinity if a complex with appropriate pyrophosphate has been already formed. As with the α subunits of other prenyltransferases, PTAR1 exhibits a crescent shape consisting of a series of right-handed antiparallel coils that form six α-helical structures surrounding a spherical β subunit. RabGGTB has the same α–α barrel motif as in RGGT, consisting of six parallel internal and six circumferential α helices. The enzyme’s active site in the cavity of the funnel-shaped structure is filled with conserved hydrophobic residues and has a zinc ion bound at its edge. The architecture of GGTase-III is very similar to the known prenyltransferase complexes [9]. 

The crystal structure analysis of the GGTase-III:FBXL2:Skp1 complex revealed an extensive multivalent interface specifically formed between the LRR domain of FBXL2 and PTAR1, which unmasks the structural basis of the substrate-enzyme specificity. The LRR domain of FBXL2 contains thirteen LRR motifs (LRR1–LRR13) and an additional β-strand followed by the C-terminus containing the CAAX sequence. The extra β-strand at the C-terminus of the protein with respect to the LRR13 creates a unique cavity termed the “LRR13 pocket”, which is involved in enzyme binding. GGTase-III interacts with the substrate primarily through the unique N-terminal extension (NTE) of PTAR1, which forms an α/β subdomain anchored to the concave surface of the FBXL2 LRR domain. The electron density for the C-terminus of FBXL2 could not be found, which supports the thesis that recognizing the F-box protein by GGTase-III is mediated by regions other than the CAAX motif of the substrate. The GGTase III:FBXL2 complex has more than 2500 Å^2^ solvent-accessible surface area, which is greater than any other prenyltransferase:substrate complex. The high-affinity binding between GGTase-III and FBXL2 is most likely due to the large intermolecular interaction area [9].

A structural feature that distinguishes PTAR1 from the other two α subunits of prenyltransferases is the presence of a highly conserved element of NTE. In contrast to the NTE, FNTA, and RabGGTA, which are largely disordered, the structure of the NTE in PTAR1 begins with the α1 helix, followed by a β sheet of three β strands (β1–β3), and ends with the α2 helix directed perpendicular to the α1 helix and connecting the NTE to the domain of the helical motifs. PTAR1 NTE is distinguished by the presence of a 10-amino-acid loop surrounded by β1 and β2 strands (β1–β2 loop). This loop, along with the rest of the NTE element, plays a key role in the recruitment of FBXL2.

Recognition of FBXL2 by GGTase-III is exclusively mediated by the α subunit of PTAR1, which covers the entire concave part of the F-box protein. In the C-terminal half of the LRR domain of FBXL2, NTE PTAR1 forms a continuous and highly complementary interaction surface with the F-box protein. The characteristic feature of this surface is the interlocking coupling between the LRR13 FBXL2 pocket and the α1 helix and the β1–β2 NTE PTAR1 loop. The top of the β1–β2 loop contains an asparagine residue, Asn43α, through which it attaches to the FBXL2 pocket, positioning itself in its depth, and forming two hydrogen bonds with the FBXL2. As a result, the β1–β2 loop and the nearby α1 helix clamp the short FBXL2 loop of three amino acid residues (Ala398, Tyr399, Phe400) and terminate an additional C-terminal β strand to form a tangled protein structure. The Phe400 closes the LRR13 pocket and secures the Asn43α PTAR1 docking inside the cavity. The FBXL2:PTAR1 interactions are further enhanced by a network of polar interactions between the α1 helix of PTAR1 and the three FBXL2 LRRs preceding the LRR13. All these features indicate a key role in mediating the binding of FBXL2 to PTAR1.

The N-terminal LRRs of FBXL2 are also involved in the interaction with PTAR1. Interactions occur between the vertices of the first five LRRs in FBXL2 and the PTAR1 loop connecting the α2 and α3 helices. In contrast to the C-terminal interface, the N-terminal region is characterized by two pairs of Trp–Arg interactions in which the aliphatic side chain of an arginine residue (Arg86 with FBXL2 and Arg84 with PTAR1) aligns with the indole ring of a tryptophan residue (Trp81 with PTAR1 and Trp165 from FBXL2). These interactions are further stabilized by a salt bridge formed between Arg190 FBXL2 and Asp85 PTAR1 [9].

The Golgi SNARE protein Ykt6 was also identified as a substrate of GGTase-III. Ykt6 belongs to the SNARE family of proteins that regulate vesicular transport within the Golgi apparatus and other transport pathways, including autophagosome–vacuole/lysosome fusion. Human Ykt6 is a 198-amino-acid protein consisting of an N-terminal longin domain and a C-terminal SNARE domain. Unlike most SNARE proteins, Ykt6 lacks a transmembrane domain. Instead, it contains a prenylation motif of two conserved cysteines at the C-terminus (C194C195AIM). The second cysteine (Cys195), which forms the CAAX sequence motif, is farnesylated by FTase. GGTase-III recognizes Cys195 farnesylated by FTase and attaches a geranylgeranyl group to Cys194, generating doubly prenylated Ykt6. Cys194 geranylgeranylation is much more efficient if the -AIM sequence is cleaved after the first prenylation step. The released carboxyl group is methylated with RCE1 and ICMT, as is the case with the other CAAX motif proteins. Besides Rab proteins, Ykt6 is the first example of a protein modified with two prenyl groups. Double prenylation of Ykt6 is essential for the proper functioning of the Golgi SNARE protein. The lack of these modifications leads to structural disorganization of the Golgi apparatus and disturbed transport of proteins inside the organelle [8].

Although the Ykt6 protein is modified with lipid residues, it is located mainly in the cytosol and may circulate between the cytosol and membranes as a result of activating and inactivating conformational changes [46]. Structural studies of rat Ykt6 in complex with dodecylphosphocholine (DPC) showed that the cause of the cytosolic localization of the protein was the adoption of a closed, autoinhibited conformation with the SNARE motif wrapped around the longin domain. The rest of the lipid is bound and isolated in the hydrophobic groove formed between the longin domain and the SNARE motif, stabilizing the closed structure of the protein [47]. The double prenylated recombinant Ykt6 showed similar properties to the monofarnesylated form of the protein. This suggests that the second prenyl group may also be masked in the hydrophobic groove and shielded from the aquatic environment [8].

In addition to the structure of apo-GGTase-III (PDB: 6J6X), the enzyme’s crystal structure in the complex with Ykt6 was also solved. The C-terminus of the PTAR1 was C-terminally truncated to facilitate crystallization. Using the Cys195–farnesyl–Ykt6 protein lacking the C-terminal sequence -AIM, it was possible to determine the structure of the protein–enzyme complex and GGPP (PDB: 6J7F, Figure 8C) in which the two terminal Ykt6 cysteines were bound to the GGTase-III active site. The complex formed represents a catalyzed state since the geranylgeranyl moiety of GGPP has been transferred to Cys194. The Ykt6 protein is recognized directly by GGTase-III and, more specifically, its α subunit. In the complex, Ykt6 binds to PTAR1 in a closed conformation, in which the SNARE domain forms three separated α helices and surrounds the longin domain, similar to what is observed in the DPC-bound Ykt6 structure. Two binding sites have been identified between PTAR1 and Ykt6. In the first one, the unique N-terminal domain of PTAR1 forms a hydrophobic pocket consisting of Ile35α, Pro36α, Val48α, and Val50α, in which Phe30 and Phe31 Ykt6 are located. The side chain of Glu31 in PTAR1 is involved in forming a hydrogen bond with the -NH group of the Phe30 Ykt6 main chain. In the second site, the interaction of PTAR1 with Ykt6 is formed by loops between α helices α9 and α10 as well as α11 and α12. The -CO group of the main chain of Met229 PTAR1 participates in forming a hydrogen bond with the amino terminal group of Met1 Ykt6. Ser232 in PTAR1 is involved in forming a hydrogen bond with the carboxyl group of the Glu84 Ykt6 side chain. In addition, Tyr306 in PTAR1 is housed in a hydrophobic pocket formed by Met1, Pro86, Pro133, and the aliphatic portion of Arg134 of Ykt6. Val231 in PTAR1 participates in hydrophobic interactions with Met1 and the aliphatic part of Glu84 in Ykt6.

The C-terminal fragment of Ykt6 (Thr187-Cys195) is placed in the central cavity of the β subunit of the enzyme. The geranylgeranyl moiety is located in the pocket that binds the lipid substrate to allow the formation of a covalent bond between the C1 carbon of GGPP and the sulfur atom Cys194. Geranylgeranylated Cys194 is located at the catalytic site formed by zinc, coordinating conserved β subunit residues Asp238β, Cys240β, and His290β. The location of Cys194 suggests that GGTase-III exhibits the same catalytic mechanism as FTase and GGTase I.

The farnesyl moiety bound to Cys195 is anchored in a hydrophobic tunnel formed near the active site. It corresponds to the exit groove of FTase and GGTase-I. In both enzymes, this tunnel is blocked by a branched side chain of the Leu103β and Ile50β of the β subunits, respectively. In GGTase-III, these residues are replaced by Gly49β, allowing the insertion of a prenyl group. Replacing Gly49β with Leu or Ile blocked the transfer of the second prenyl group to mono-farnesylated Ykt6. Thus, the unique RabGGTB tunnel plays a key role in transferring the second prenyl residue. The anchorage of the farnesyl group bound to Cys195 in the tunnel allows the correct location of the unmodified Cys194 in the active site. The above structural findings are in line with the results that GGTase-III preferentially modifies C195–farnesyl–Ykt6. After the geranylgeranylation is completed, a new GGPP molecule attachment to the enzyme releases the modified product. Its isoprenoid residues are transferred to the prenyl-binding pocket in the Ykt6 protein, analogous to the situation in the REP protein after the double geranylgeranylation of Rab proteins by RGGT. Gly49β in RGGT is particularly important in the relocation of various C-terminal variants of the monoprenylated protein sequence, which cannot form well-defined interactions with the enzyme active site. The Gly49β residue enables the correct placement of the second unmodified cysteine residue in the active center of RGGT [8].

## 6. Selected Inhibitors of Prenyltransferases

Inhibitors of prenyltransferases commonly interact with these enzymes via coordination of the zinc ion, the lipid-binding and/or CAAX substrate-binding site, the exit groove, or specific to a particular enzyme-binding site. They are often classified as (1) analogs competitive with the isoprenoid substrate, (2) compounds that are competitive with the protein substrate, (3) bisubstrate inhibitors, which bear both of the features mentioned above, and (4) compounds that are neither competitive with the protein nor the isoprenoid substrate [17]. The unfavorable features of prenyl pyrophosphate, one of the substrates for this class of enzymes, include its anionic character and promiscuity; as natural pyrophosphates, they have a high affinity to other enzymes, which makes them less attractive as molecules with potential therapeutic properties. This review discusses only selected inhibitors and their interactions with prenyltransferases. The inhibitors were chosen based on the availability of the crystal structures of their complexes with enzymes or their high potency.

### 6.1. FTase Infibitors (FTIs)

Several peptidomimetic FTIs were developed, which are competitive towards peptide substrates, either by blocking fully or partially the peptide site and the exit groove, where the displaced prenyl group of lipidated product binds (Table 2) [48]. First efforts focused on converting the native K-Ras CAAX box, bearing the CVIM sequence, into FTIs. While thiol analogs show activity against FTase (e.g., L-739749), the thiol group is associated with toxicity and instability (proneness to oxidation). Therefore, non-thiol analogs were designed. To avoid losing coordination to Zn^2+^, the thiol group was exchanged for imidazole, another group with Zn^2+^-coordinating properties. Such molecular evolution led to a more drug-like molecule, BMS-214662, competing with the CAAX substrate for FTase [49] (Figure 9A).

On the other hand, high-throughput screening led to the discovery of potent FTIs such as tipifarnib (R11577) (Figure 9B) and lonafarnib (SCH66336), which progressed to Phase III clinical trials, with the latter, under the name Zokinvy™, being approved in the USA for Hutchinson-Gilford Progeria Syndrome (HGPS) and processing-deficient progeroid laminopathies (Table 2) [50]. Both turned out to be peptide-substrate-competitive inhibitors [51,52]. Although most FTIs bind the zinc ion, lonafarnib does not, but it blocks the peptide site and exit groove. Its pyridinyl ring contributes to strong stacking interactions with Tyr361β, and the other six-membered rings of the tricycle interact with Tyr166α and His201α.

**Figure 9 ijms-23-05424-f009:**
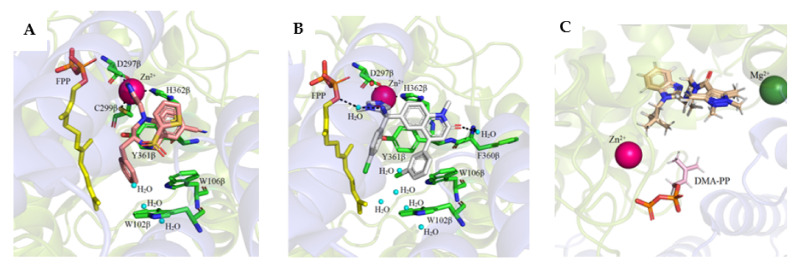
Structures of farnesyltransferase complexed with FPP and BMS-214662 (**A**) (PDB: 1SA5), with FPP and R115777 (**B**) (PDB: 1SA4), and with compound **1** [53] (**C**) (PDB: 7RNI). (Created in the PyMOL Molecular Graphics System, Version 2.5.2 Schrödinger, LLC).

The non-thiol-containing FTI-2148 is a selective inhibitor of FTase (Table 2) [54]. The crystal structure of the FTase:FPP:FTI-2148 complex shows that the inhibitor has a similar conformation to CVIM. The imidazole coordinates the zinc ion, and the carboxylic acid (from methionine) and carbonyl from the amide bond form two hydrogen bonds with Gln167α and Arg202β, respectively. At the same time, the biphenyl interacts in a hydrophobic pocket with aromatic residues of Trp102β, Trp106β, and Tyr361β [55].

In the recently conducted studies, a new FTI analog, compound **1**, was identified in a high-throughput phenotypical screen as a molecule that potentiates the activity of histone deacetylase inhibitor, vorinostat, to reverse HIV latency (Table 2, Figure 9C) [53]. Several diverse techniques were used to determine that compound **1** is targeting FTase. Interestingly, under standard conditions of biochemical FTase assay, no activity of **1** was observed. Rather, **1** inhibited FTase in a purified enzyme assay only in the presence of polyvalent anions, which likely play a role as a surrogate for diphosphate in FPP, as has been previously observed [56]. The X-ray structure of the complex formed between **1** and FTase, in the presence of truncated pyrophosphates, such as dimethylallyl diphosphate (DMA-PP) (PDB: 7RNI, Figure 9C) or pyrophosphate (PPV), shows two hydrogen bonds between **1** and the side chain of Ser99β and Arg202β. The binding mode for **1** is analogous to that observed for lonafarnib (PDB: 1O5M) [52], showing that **1** interacts with the FTase active site via partially occupying both the FPP and CAAX pockets. The neopentyl group is probably responsible for essential binding in a hydrophobic pocket since an analog with a shorter ethyl chain lacks activity [53]. 

In the recent studies using only virtual HTS of phyto compounds, ascorbic acid was identified as a possible inhibitor of FTase. The 3D-QSAR studies indicate that ascorbic acid interacts via hydrogen bonds with several residues in the active site of FTase (Lys353, Tyr300, Leu295, Gly290) [57].

**Table 2 ijms-23-05424-t002:** Selected inhibitors of FTase.

Entry	Inhibitor	Potency	PDB CODE	Mode of Binding *	Reference
1	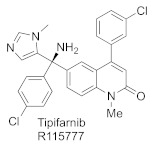	IC_50_ 0.86 nM (FTase); 90 clinical trials	1SA4	protein substrate competitive	[58]
2	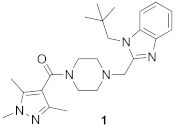	EC_50_ 0.34 μM @ EC 10 Vorinostat EC_50_ > 8 μM without Vorinostat	7RNI	protein substrate competitive	[53]
3	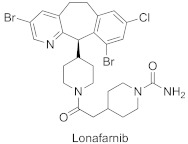	IC_50_ 1.9 nM (FTase); 40 clinical trials	1O5M	protein substrate competitive **	[52]
4	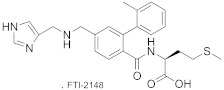	IC_50_ 1.4 nM (FTase); IC_50_ 1.7 μM (GGTase-I)	=	-	[54]

* Mode of binding given only for enzyme–ligand complexes solved by crystal structure. ** Partially occupy the exit groove, overlapping with the displaced product farnesyl moiety.

### 6.2. GGTase-I Inhibitors (GGTIs)

Most GGTIs are protein-substrate-competitive inhibitors, with the structures derived from the tetrapeptide of C-terminal sequence CAAX (Table 3). However, FTase inhibitors tend to be smaller and more hydrophilic than GGTIs [48]. Therefore, to obtain selective GGTIs, the peptide backbone should be modified with more hydrophobic elements. Among GGTIs, thiol and non-thiol-containing analogs can be distinguished, such as GGTI-287 [59], GGTI-2133 (IC_50_ value of 38 nM toward GGTase-I) [60], and GGTI-2154 [54]. Exchange of a thiol for an imidazole in GGTI-287 led to GGTI-2154, an inhibitor with over 250-fold selectivity for GGTase-I over FTase [54], while further tuning led to GGTI-2418, an inhibitor with over 5000-fold selectivity for GGTase I over FTase [61]. GGTI-2418 is the only GGTase-I inhibitor that progressed to clinical trials (IC_50_ = 9.5 nM) [62], showing a good safety profile and fast elimination properties. In turn, GGTI-DU40 is a selective non-peptidomimetic inhibitor of GGTase-I (IC_50_ 8.24 nM), isolated in optimization studies of a GGTI selected from a library of chemically diverse compounds. Therefore, it is structurally distinct from other inhibitors. Furthermore, its mode of inhibition is competitive towards the CAAX peptide-binding site and uncompetitive toward GGPP [63]. 

GGTase-I inhibitors bearing different core structures, the tetrahydropyridine and the dihydropyrrole ring (e.g., P3-E5 and P5-H6), are not characterized by X-ray, but they probably compete with the substrate protein [64]. Their analogs, compounds P61-A6 and P61-E7 (the latter derived from P3-E5 by addition of L-Leucine methyl ester), retain potency against GGTase-I, showing activity in different cancer models [65,66,67].

**Table 3 ijms-23-05424-t003:** Selected inhibitors of GGTase-I.

Entry	Inhibitor	Potency	PDB CODE	Reference
1	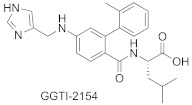	IC_50_ 21 nM (GGTase-I) IC_50_ 5.6μM (FTase)	-	[54]
2	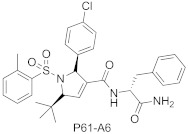	IC_50_ 2.2 μM (GGTase-I) (K562 Proliferation)	-	[65]
3	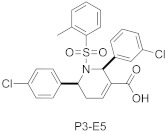	IC_50_ 313 nM (GGTase-I)	-	[64]
4	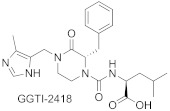	IC_50_ 9.5 nM (GGTase-I) IC_50_ 53 μM (FTase) (recruiting for clinical trial)	-	[62]
5	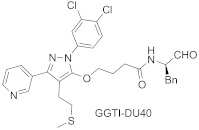	IC_50_ 8.24 nM(GGTase-I) IC_50_ > 2 μM(FTase)	-	[61]

### 6.3. Selected Dual Inhibitors of FTase and GGTase-I

Despite many clinical trials of FTase inhibitors, only lonafarnib and tipifarnib have advanced to Phase III clinical trials. Several reasons explain this failure, including the possibility of surrogate prenylation by GGTase-I of one of the most commonly mutated Ras oncoproteins, K-Ras, and the selection of patients with advanced and/or metastatic disease [10]. Therefore, agents inhibiting both FTase and GGTase-I are also considered therapeutically relevant. FGTI-2734 is a CAAX motif-derived peptidomimetic inhibitor of equal potency toward farnesyl and geranylgeranyl transferases (Table 4). It turned out to be effective in several models, showing the inhibition of K-Ras membrane localization in several lines of human cancer cells and suppressing a few other oncogenic pathways. It also inhibits the membrane localization of N-Ras, having potential applicability for N-Ras-driven cancers [68].

L-778,123 is a dual inhibitor of both FTase and GGTase-I, but the interaction mechanisms with each of these enzymes are different [58]. According to the ternary complex with FTase/FPP (Figure 10A; PDB: 1S63), L-778,123 is a peptide-competitive inhibitor, mimicking the CA_1_A_2_ part of the peptide substrate. The imidazole group coordinates the zinc ion and the rest of L-778,123 forms van der Waals interactions, including interactions between the *p*-cyano-phenyl ring and Tyr166α and the FPP farnesyl moiety, as well as interactions between piperazine-2-one and *m*-chloro-phenyl rings and Trp102β, Trp106β, and Tyr361β. In the complex with GGTase-I, the inhibitor occupies both the lipid-binding pocket (overlapping with the binding of GGPP isoprenes 1-3) and a portion of the peptide-substrate-binding pocket, specifically C and A_2_ from the CA_1_A_2_X peptide. The remaining GGTase-I residues, responsible for interaction with the fourth isoprene unit and X amino acid from the CAAX box, are not in contact with the inhibitor (Figure 10B; PDB: 1S64). L-778,123 is an anion-dependent inhibitor since the complex formation requires the presence of an anion (e.g., sulfate, phosphate), which is bound to the pocket, in the position of a phosphate group from GGPP. In both complexes, with FTase and GGTase-I, L-778,123 adopts the U-shaped turn and the imidazole nitrogen coordinates the zinc cation, simultaneously overlapping with the cysteine of the CVIL-GGTase-I complex. The crystal structure of the complex of low-affinity GGTase-I/GGPP/L-778,123 was also acquired, in the absence of sulfate or phosphate anions. It is consistent with a peptide-competitive mode of inhibitor binding, with two molecules of GGPP. The piperazine-2-one group interacts with the enzyme within the CA_1_A_2_X protein-substrate-binding site, in the position that overlaps with valine and isoleucine in the crystal structure of the CVIL-GGTase-I complex (PDB: 1N4Q vs. PDB: 1S64).

To disrupt electrostatic protein–protein interactions between K-Ras and two prenyltransferases, FTase and GGTase I, bivalent inhibitors were obtained, derived from the known peptidomimetic FTI-277, which was combined with a guanidyl-containing gallate moiety responsible for interaction with the common acidic surface of FTase and GGTase I. Such a compound, in a cell-based assay, proved to disrupt the localization of K-Ras to the plasma membrane and impair the interaction with c-Raf [69].

### 6.4. GGTase-II Inhibitors

All prenyltransferases share some similarities, leading to problems with developing selective inhibitors. While searching for distinguishing properties between GGTase-II (RGGT, Rab GGTase) and two other prenyltransferases, FTase and GGTase I, a number of features were identified, such as the REP protein, which is mainly responsible for the recognition and delivery of protein substrate Rab GTPases to RGGT and the larger binding cavity in RGGT compared with other prenyltransferases. The most recent distinction involves the identification of the two sites—the tunnel adjacent to the GGPP binding site, the so-called TAG tunnel, together with the lipid-binding site (LBS)—both of which need to be simultaneously targeted to achieve selective binding to RGGT [70,71].

A TAG tunnel was identified during the screening of a library of 469 peptide-based compounds derived from the farnesyl transferase inhibitor pepticinnamin E [72]. Several compounds were identified as selective micromolar inhibitors of GGTase-II (such as compounds **2-5**, Table 5). Analysis of the interactions in crystal structures led to the identification of the TAG tunnel, a unique feature of Rab GGTase, opening the possibility of developing selective RGGT inhibitors by targeting this site. These selective RGGT inhibitors bind in a competitive or partially competitive mode towards prenyl pyrophosphate [72].

BMS3 is a highly potent dual inhibitor of Rab GGTase (Figure 11A; PDB: 3PZ3) and FTase (Figure 11B; PDB: 3PZ2). The selectivity with respect to GGTase-I is achieved thanks to the tetrahydrobenzodiazepine core (THB), which forms favorable van der Waals interactions with aromatic residues present in the Ftase and GGTase-II active sites, which are absent in GGTase-I [70]. The crystal structures of complexes RabGGTase: BMS3, RabGGTase: BMS3:GGPP, and Ftase: BMS3:FPP (PDBs: 3PZ1,3PZ2,3PZ4, respectively) were solved, facilitating the design of more selective Rab GGTase inhibitors. The binding modes of BMS3 with these two enzymes are similar and, among other features, include imidazole coordinating Zn^2+^, the π-stacking (the phenyl ring of THB with the Tyr361 or the Phe289 residue), and the T stacking (3-benzyl group with hydrophobic lipid-binding sites) interactions [72].

Based on these structural studies, the modifications were introduced into BMS3 to increase its potency against Rab GGTase, not Ftase [70,71]. The anisylsulfonyl residue was retained as it does not significantly influence binding with Ftase, while forming a hydrogen bond to Tyr44 in Rab GGTase. Two positions in BMS3 were considered crucial for further optimization towards increasing selective binding with Rab GGTase, thanks to increasing affinity for two sites, the TAG tunnel and lipid-binding site (LBS). The nitrile group could be extended to fit into the TAG tunnel while clashing with the Ftase surface. The second moiety, the 3-benzyl group, which interacts with the LBSs of both enzymes, could be larger, as this cavity is larger in Rab GGTase than in Ftase. Such structural analysis was supported by virtual high-throughput screening [70].

While the modification at the site interacting with the TAG tunnel affected the inhibitory activity against Rab GGTase, the fully selective analogs were obtained only upon the modification at both TAG and LBS sites, giving compounds **6** and **7** (Table 5). The analysis of the crystal structure of the complex RabGGTase: **6** (PDB: 3PZ3) shows that the THB core π stacks with Phe289, the benzyl carbamate is localized in the lipophilic substrate-binding site, and T stacking stabilizes this interaction (e.g., with Phe147). The furanal part is at the entrance of the TAG tunnel, but no hydrogen bonding between aldehyde and the Tyr30 moiety is observed [70]. These hits were confirmed in further studies of the same group [71].

The natural product psoromic acid (PA) (Table 5) was extracted from a library of 13,030 compounds as a potent and selective inhibitor of Rab GGTase. Compared with other RGGT inhibitors, PA has a different mode of interaction. While THB and peptide inhibitors coordinate the zinc ion and achieve selectivity by targeting the TAG tunnel, PA does not associate with these sites. Based on SAR studies, the depsidone core structure and a 3-hydroxyl and 4-aldehyde motif were identified as crucial for the selective inhibition of Rab GGTase. According to the crystal structure, the PA interacts with the isoprenoid binding site in a competitive mode, probably with phenyl ring B of PA (the one without an aldehyde group) occupying the location of the second isoprenyl unit of GGPP, and two phenyl rings interacting with Trp244β and Phe289β (Figure 11C; PDB: 4EHM). The lactone carbonyl forms a weak hydrogen bond with Trp52β via a water molecule. In the crystal structure of this complex, the hydrogen bonding between the N-terminal region of the α subunit with the β subunits was observed. PA forms a covalent imine bond with the N-terminus of the α subunit, which seems to enhance the interaction of PA with the active site. The crystal structure of the RGGT-PA complex was determined with the use of a reducing agent, NaCNBH_3_, which transformed the labile imine bond into a stable amine bond. PA is not coordinating the zinc ion. Rab GGTase is autoinhibited by the coordination of N-terminal His2α with the zinc ion. Such autoinhibition was not observed for other prenyltransferases, FTase and GGTase-I, which do not contain an analogous αHis2 residue and are faster than RGGT [73].

Another class of micromolar inhibitors of GGTase-II constitutes compounds derived from GGTase-I inhibitors, bearing a pyrrolidine ring. The unique feature, responsible for the selectivity of these compounds towards RGGT, compared with GGTase-I, is the hydrophobic chain connected with C4 in the pyrrolidine ring via a thioether bond. This class of inhibitors competes with the substrate protein [64], and the most potent among them is the compound P49-F6.

Phosphonocarboxylates (PCs) constitute another class of GGTase-II inhibitors. A thorough SAR analysis was carried out for PCs, including the synthesis of non-covalent and covalent analogs. It was found that the most potent phosphonocarboxylate analogs bear an imidazole (compounds **8**, **9**) [74] or imidazo[1,2-*a*]pyridine ring (3-IPEHPC and compounds **10-11**). In the latter case, only substitution in the 6 position of the heterocyclic ring gives an RGGT inhibitor [75,76]. Thorough studies were performed on a representative of this class, (+)-3-IPEHPC [77]. Based on these studies, it was concluded that (+)-3-IPEHPC probably binds to the site distinct from the protein substrate and GGPP-binding sites on RGGT, probably interfering with the complex rearrangement that must occur in order to enable the second geranylgeranylation. Based on docking studies, it was suggested that the α-phosphonocarboxylate moiety guides the location of the inhibitor by stabilization via hydrogen bonding interactions and coordination of the zinc ion to phosphonic acid [75]. The substituent in position 6 faces the indole side chain in Trp244β, giving favorable packing, while His190β and Tyr107α might stabilize the conformation by hydrogen bonding with N1 in imidazo[1,2-*a*]pyridine. It was found experimentally on the example of 3-IPEHPC [78] that there is a preference for the binding of (*S*)-enantiomer, as this stereoisomer shows higher potency compared with (*R*)-3-IPEHPC. As for 6-Me-F-MinPC, the difference in potency between two stereoisomers was also observed, and it was suggested that one configuration is given a preference for binding within the active site. Docking studies show that in the (*S*)-isomer, the fluorine interacts with Cys240β and/or Trp244β, while such interaction is not observed for fluorine in (*R*)-enantiomer [75,78].

Phosphonocarboxylates equipped with an electrophilic group were used to identify the site of interaction with RGGT [79]. For this purpose, mass spectrometric analysis and isotopically labeled probes were applied [80], and the studies included an in-depth analysis of the applicability of diverse digesting enzymes. One analog, compound **11**, turned out to be an inhibitor of RGGT, potentially having a covalent mode of action. Cys196β or/and Cys197β were identified as the cysteines targeted by the electrophilic moiety. These cysteines are not within the TAG tunnel and the hydrophobic part of the electrophilic moiety probably interacts with the Trp244β indole side chain.

**Table 5 ijms-23-05424-t005:** Selected inhibitors of Rab GGTase.

Entry	Inhibitor	Potency	PDB CODE	Mode of Binding *	Reference
1	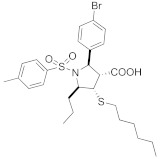	IC_50_ 2.1 μM	-		[64]
9	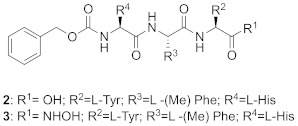	**2:** IC_50_ 22.7 ± 1.7 μM (Rab GGTase) **3:** IC_50_ 9 ± 1 μM (Rab GGTase)	**2:** 3C72 **3:** 3HXD	TAG tunel	[72]
	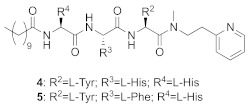	**4:** IC_50_ 11 ± 1.2 μM (Rab GGTase) **5:** IC_50_ 4.7 ± 0.1 μM (Rab GGTase)	**4:** 3HXF **5:** 3HXE	TAG tunel	[72]
2	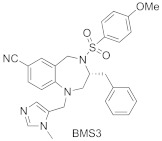	IC_50_ 6.4 ± 4.8 nM (Rab GGTase)^RA^ IC_50_ 724 ± 321 nM (Rab GGTase)^FL^	3PZ1 (BMS3:RGGT), 3PZ2 (BMS3:RGGT:GGPP), 3PZ4 (BMS3:FTase:FPP)		[70,71]
3	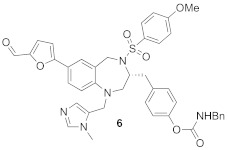	IC_50_ 1547 ± 101 nM (Rab GGTase)^RA^ IC_50_ 616.2 ± 415.7 nM (Rab GGTase)^FL^	3PZ3	TAG tunel and LBS	[70,71]
2	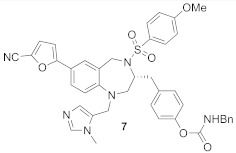	IC_50_ 260 ± 18 nM (Rab GGTase)^RA^ IC_50_ 41.6 ± 9 nM nM (Rab GGTase)^FL^	-	TAG tunel and LBS	[70,71]
3	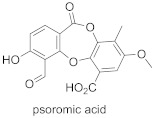	IC_50_ 1.4 μM	4EHM	Isoprenoid substrate competitive	[73]
4	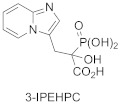	LED 25 μM	-		[78]
5	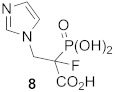	LED 10 μM	-		[81]
6	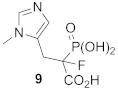	LED 10 μM	-		[74]
7	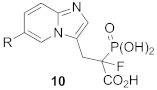	R = Me, LED 10 μM R = Br, LED 10 μM R = p-CHO-C_6_H_4_, LED 25 μM	-		[75,76]
8	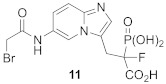	IC_50_ 154 μM LED 25 μM	-		[79]

* Mode of binding given only for enzyme–ligand complexes solved by crystal structure.

## 7. Conclusions

Protein prenylation has been studied extensively for nearly 30 years and the structural characterization of mammalian prenyltransferases has improved our understanding of these enzymes. In healthy cells, prenylation of the Ras superfamily GTPases is necessary for diverse cellular processes, such as growth, cell movement, vesicular transport with sorting the cargos, the formation and transport of vesicles, and their fusion with the target membranes. Prenylation serves as the first critical step for membrane targeting and binding, as well as mediating protein–protein interactions [82]. Dysregulation of the activity of small GTPases has been related to a variety of disorders due to their critical roles in the cells. The Ras protooncogenes H-Ras, K-Ras, and N-Ras, which are mutated in approximately 25% of malignancies, are the most well-known instances [83]. The search for new inhibitors of prenylating enzymes is justified, as the deregulation or deregulation of their substrates’ activity is broadly associated with a number of diseases [1]. Most of these studies concentrate on targeting diverse malignancies, such as breast, ovarian, lung, and liver cancers. However, there are also efforts at using, e.g. lonafarnib against infectious diseases, such as chronic hepatitis D. The lonafarnib has been recently approved for Hutchinson-Gilford Progeria Syndrome (HGPS) and processing-deficient progeroid laminopathies. A number of clinical studies were run, mostly focused on the use of FTase inhibitors. Their initial failure resulted from a lack of knowledge about the possibility of surrogate prenylation of some Ras proteins. Therefore, currently, the regime of choosing the candidate patients for clinical trials is more rigorous. For example, some tipifarnib studies include targeting H-Ras mutation in different cancers, as in patients with K-Ras mutation, the treatment would be less efficient, due to possible protein prenylation mediated by GGTase-I. 

The significance of structural X-ray studies in this effort cannot be overestimated and this technique still constitutes the main tool for designing potent, selective inhibitors of a particular enzyme. For example, based on structural data, it can be concluded that FTIs are typically smaller than GGTIs and a hydrophilic molecule’s binding affinity for FTase is commonly stronger than GGTase-I. Therefore, adding a polar group to a hydrophilic molecule would be a viable choice for designing a selective FTase inhibitor [48]. Taking into consideration the structural overlap with other enzymes of this class, several cases were reported (some discussed above), when structural data enabled the derivation from an inhibitor targeting two prenylating enzymes, a selective molecule, thanks to the presence of a specific binding site (such as the TAG tunnel in RGGT). Such data also help in building the libraries of compounds for virtual screenings, having constraints imposed by X-ray structures. However, as these approaches can be extended for designing inhibitors of the most recently reported prenyltransferase, GGTase-III, new challenges can be spotted in this area and the rising significance of complementary techniques, such as mass spectrometry studies, should be acknowledged. Moreover, novel approaches are being proposed for the direct control of GTPases [83], via blocking some protein–protein interactions, including interference with regulatory proteins. This approach is especially valuable when it comes to Rab proteins, which constitute the largest class of prenylome, and the selective targeting of a member or a few members of this protein class cannot be achieved by the inhibition of GGTase-II.

## Figures and Tables

**Figure 1 ijms-23-05424-f001:**
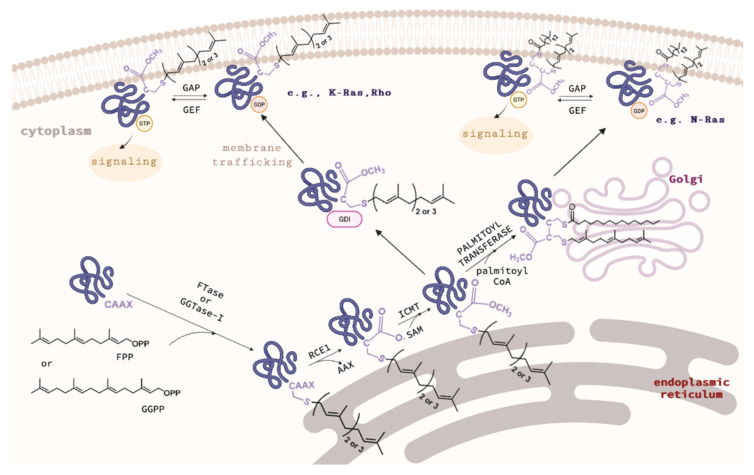
Processing pathway of CAAX prenyltransferases (created in BioRender.com). The CAAX proteins are synthesized in a soluble form and then modified by protein prenyltransferase, FTase, or GGTase-I. Further processing by RCE1 and ICMT generates mature forms of S-prenylated proteins targeted and anchored to the appropriate membranes. In some cases, S-palmitoylation is necessary to immobilize proteins in membranes stably.

**Figure 2 ijms-23-05424-f002:**
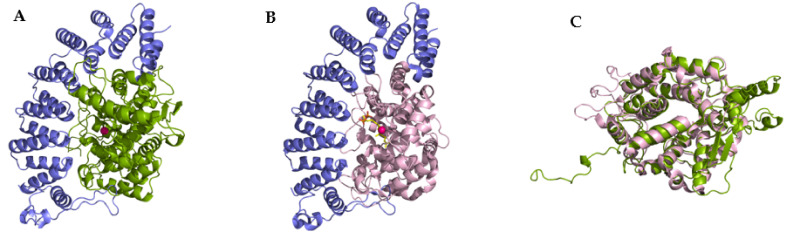
Overall structures of FTase (**A**) (PDB: 1FT1) and GGTase-I (**B**) (PBD: 1N4P), with the α subunit shown in blue, the FTase β subunit in green, and the GGTase-I β subunit in pink. The catalytic zinc ion is shown in red. Superposition of β subunit of FTase and GGTase-I (**C**). (Created in the PyMOL Molecular Graphics System, Version 2.5.2 Schrödinger, LLC.).

**Figure 3 ijms-23-05424-f003:**
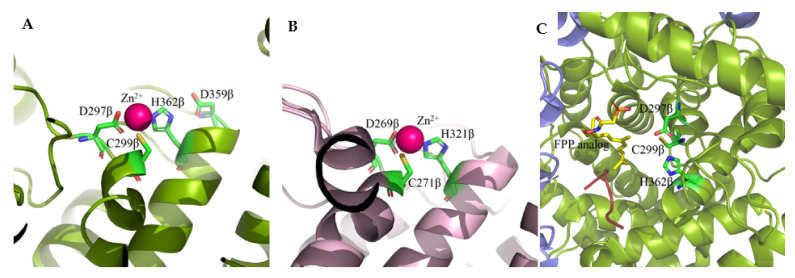
A single Zn^2+^ ion bound to the β subunit of FTase (**A**) (PDB: 1FT1) and GGTase-I (**B**) (PBD: 1N4P) is coordinated by three conserved amino acid residues, Asp, Cys, and His. Zinc-depleted FTase complexed with K-RAS4B peptide substrate and FPP analog (**C**) (PDB: 1D8E). (Created in the PyMOL Molecular Graphics System, Version 2.5.2 Schrödinger, LLC.).

**Figure 4 ijms-23-05424-f004:**
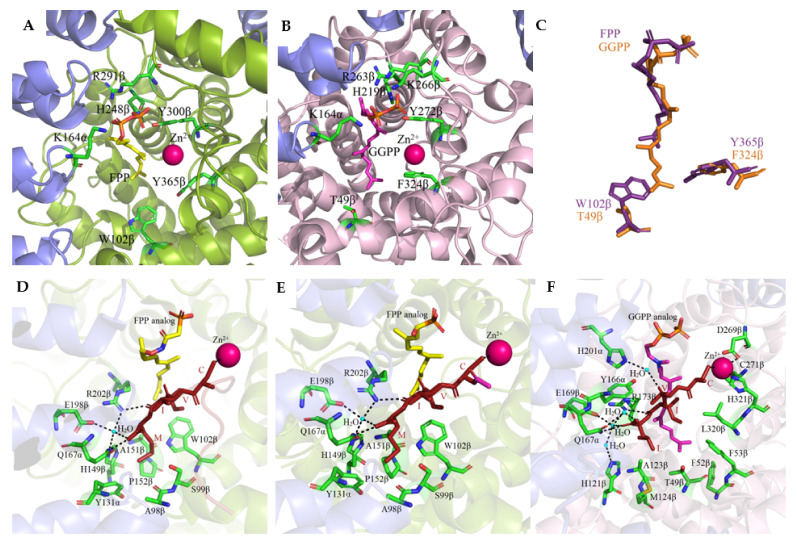
The co-crystal structure of FTase complexed with a farnesyl diphosphate substrate (**A**) (PDB: 1FT2) and with a K-Ras4B peptide substrate and FPP analog (**D**) (PDB: 1D8D) and with a CVIM peptide and FPP analog (**E**) (PDB:1QBQ) and GGTase-I complex with geranylgeranyl diphosphate (**B**) (PDB: 1N4P) and with a GGPP analog and a KKKSKTKCVIL peptide (**F**) (PDB: 1N4Q). Comparison of isoprenoid binding in FTase (PDB: 1FT2) and GGTase-I (PDB: 1N4P) (**C**). (Created in the PyMOL Molecular Graphics System, Version 2.5.2 Schrödinger, LLC.).

**Figure 5 ijms-23-05424-f005:**
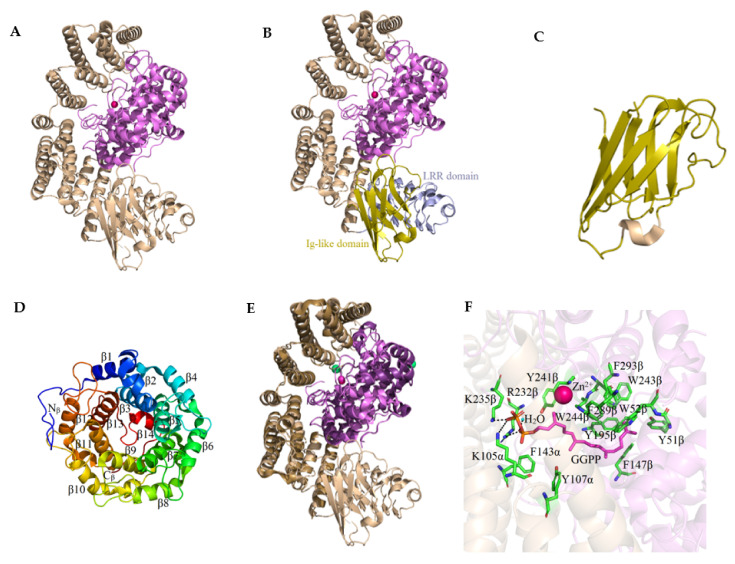
Overall structures of Rab geranylgeranyltransferase, with the α subunit shown in beige and β subunit in pink (**A**) (PDB: 1DCE). Representation of the RGGT structure, with three domains of the α subunit shown in different colors (beige, the α-helical domain; yellow, the Ig-like domain; blue, the LRR domain containing leucine-rich repeats) (**B**). The labeled strands of the Ig-like domain (**C**). The β subunit of RabGGT, and the helices numbered β1–β14 and in different colors (**D**). Superposition of 1DCE (beige, α subunit; pink, β subunit) and 3DSS (brown, α subunit; violet, β subunit) (**E**). The crystal structure of RGGT in complex with geranylgeranyl pyrophosphate (**F**) (PDB: 3DST). The zinc ion is shown in red. (Created in the PyMOL Molecular Graphics System, Version 2.5.2 Schrödinger, LLC.)

**Figure 6 ijms-23-05424-f006:**
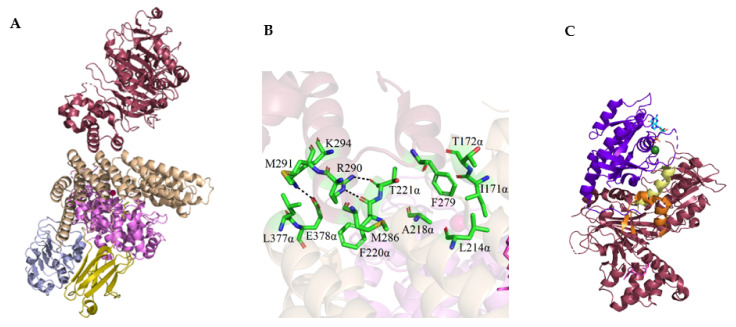
Structure of isoprenoid-bound Rab geranylgeranyl transferase complexed to REP-1 (beige, the α-helical domain; yellow, the Ig-like domain; blue, the LRR domain; pink, the β subunit; red, REP-1) (**A**) (PDB: 1LTX). The phosphoisoprenoid binding pocket of RGGT (**B**). Structure of the REP-1 (yellow, the Rab binding platform RBP; orange, the C-terminus binding region CBR) in complex with monoprenylated Rab7 (**C**) (PDB: 1VG0). (Created in the PyMOL Molecular Graphics System, Version 2.5.2 Schrödinger, LLC.)

**Figure 7 ijms-23-05424-f007:**
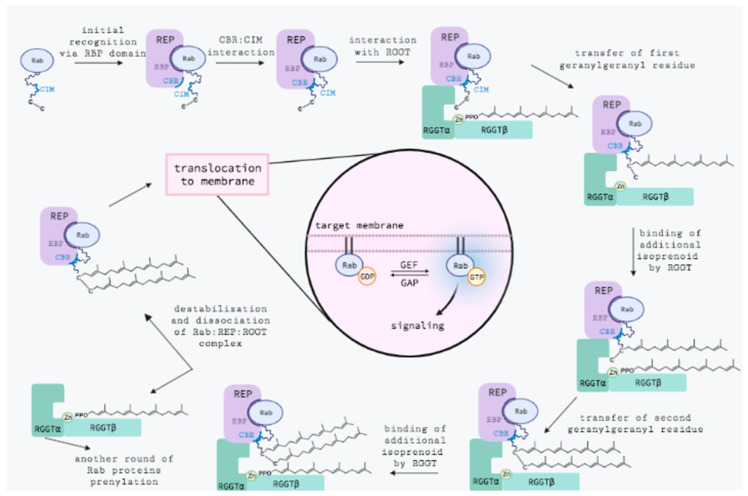
Model of RGGT-catalyzed protein prenylation mechanism (created on BioRender.com).

**Figure 8 ijms-23-05424-f008:**
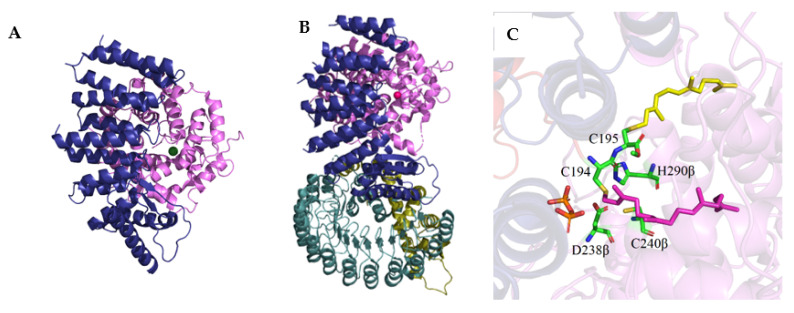
Structure of GGTase-III (**A**) (PDB: 6J6X), GGTase-III-FBXL2-SKP1 complex (**B**) (PDB: 6O60), and complex of GGTase-III, farnesyl-Ykt6 (C-terminal methylated), and GGPP (**C**) (PDB: 6J7F). (Created in the PyMOL Molecular Graphics System, Version 2.5.2 Schrödinger, LLC.)

**Figure 10 ijms-23-05424-f010:**
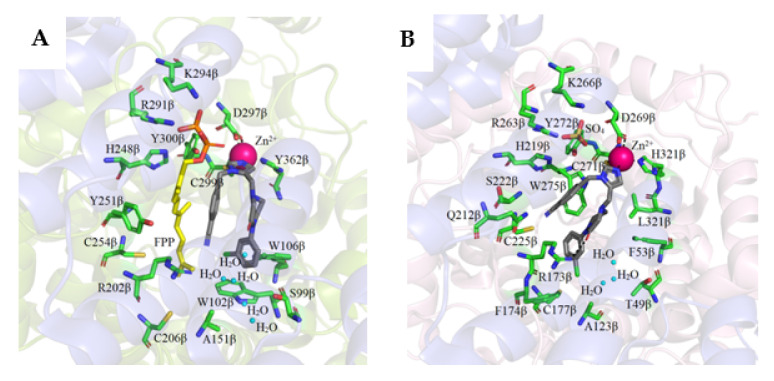
Structure of farnesyltransferase complexed with L-778,123 (**A**) (PDB: 1S63) and geranylgeranyltransferase type-I complexed with L-778,123 (**B**) (PDB: 1S64). (Created in the PyMOL Molecular Graphics System, Version 2.5.2 Schrödinger, LLC.)

**Figure 11 ijms-23-05424-f011:**
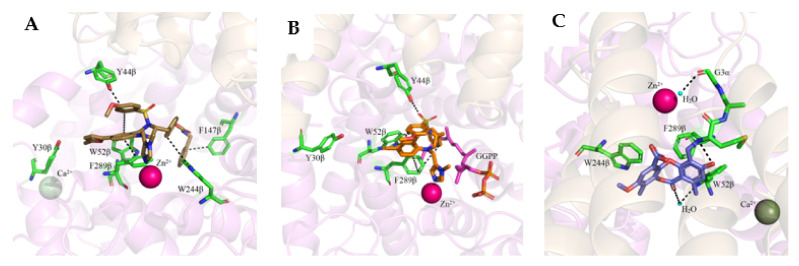
Structure of Rab GGTase with BMS3 (**A**) (PDB: 3PZ3), FTase with BMS3 (**B**) (PDB: 3PZ2), and Rab GGTase in complex with covalently bound psoromic acid (**C**) (PDB: 4EHM). (Created in the PyMOL Molecular Graphics System, Version 2.5.2 Schrödinger, LLC.)

**Table 1 ijms-23-05424-t001:** Comparison of the structure and biochemical properties of protein prenyltransferases.

	FTase	GGTase-I	RGGT (GGTase-II)	GGTase-III
**Subunit composition**	FNTA (PTAR2) and FNTB	FNTA (PTAR2) and PGGT1B	RabGGTA (PTAR3) and RabGGTB	PTAR1 and RabGGTB
**Size of subunits (mammalian)**	44 kDa and 49 kDa	44 kDa and 42 kDa	65 kDa and 37 kDa	46kDa and 37kDa
**Protein recognition sequences**	CAAX	CAAX	CC, CXC, CCX, CCXX, CCXXX (CAAX, CXXX for monogeranylgeranylation)	CAAX
**Protein substrates**	Ras, nuclear lamins	Rho, Rac, Rap	Rab	FBXL2, Ykt6
**Lipid donor substrates**	Farnesyl diphosphate (FPP)	Geranylgeranyl diphosphate (GGPP)	Geranylgeranyl diphosphate (GGPP)	Geranylgeranyl diphosphate (GGPP)
**Accessory proteins**	-	-	REP (REP-1 or REP-2)	SKP1
**Metal requirements**	Zn^2+^, Mg^2+^	Zn^2+^	Zn^2+^	Zn^2+^

**Table 4 ijms-23-05424-t004:** Selected dual inhibitors of FTase and GGTase-I.

Entry	Inhibitor	Potency	PDB CODE	Mode of Binding *	Reference
1	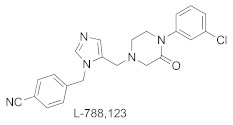	IC_50_ 2 nM (FTase); IC_50_ 98 nM (GGTase-I) (two completed clinical trials)	1S63	FTase: protein substrate; GGTase-I: protein and isoprenoid substrates competitive	[58]
2	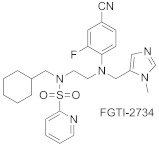	IC_50_ 250 nM (FTase) IC_50_ 520 nM(GGTase-I)	-	-	[68]

* Mode of binding given only for enzyme–ligand complexes solved by crystal structure.

## Data Availability

Not applicable.

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
