# Peer review of "Protein Prenyltransferases and Their Inhibitors: Structural and Functional Characterization"

_ijms, 2022, doi:10.3390/ijms23105424_

Round 1

Reviewer 1 Report

The review entitled "Protein prenyltransferases and their inhibitors: structural and functional characterization" by Aleksandra Marchwicka , Daria KamiÅ„ska , Mohsen Monirialamdari , Katarzyna M. BÅ‚ażewska and Edyta Gendaszewska-Darmach  describes all four human protein prenyltransferases. The article focuses on the structure of the enzymes and reviews knowledge on the known inhibitors, both the ones used in clinical practise and the ones recently discovered.

The article is comprehensive and has a very clear structure, which facilitates  reading. It contains carefully prepared tables and well thought out diagrams. The direct comparisons of the structural data of the same enzyme binding two compounds or the analogous structure in two related enzymes are very helpful for understanding the differences and similarities, that would be otherwise difficult to catch. The bibliography is adequate and contains citations of both old and very recent original works and not the reviews, and is a valuable source of information.

The general impression of this article is very good, but still there are some minor suggestions:

  • relatively long sections in the article describe the structures of protein prenyltransferases but the clinical significance of inhibiting this class of enzymes is treated rather superficially as well as the general description of the cellular/ physiological processes that are dependent on protein prenylation and hence influenced by protein prenyltranferase inhibitors. This gives the impression that a detailed structural knowledge is gathered on prenylatransferases, but the physiology of the inhibitor action is not that well studied. A suggestion to the authors is to add few sentences on this topic.
  • in the Introduction section there are some inaccuracies: not all lipid modifications of the proteins are introduced to their C-termini (line 30), it is rather specific for prenylation, while for example myrystoilation may happen close to N-termini and palmitoylation at any place in the aminoacid sequence of the protein.
  • In the same paragraph a sentence stating that three or four isoprene units may be attached to the protein (line 34) is misleading and gives the impression that the isoprene units are added to the protein and elongation of the isoprenoid is happening on the protein.
  • In the next paragraphs groups of organisms are listed: animals, plants, etc. and other species. The word species do not fit here, as previous names denote much larger evolutionary groups (line 54).
  • Also the statement that Rab GTPases are undruggable (line 64), because they do not posess small molecules binding pockets- they do, they bind guanine nucleotides.
  • line 116 - maybe few words about targeting with inhibitors ICM and STE24 enzymes as an alternative for inhibition of prenyltransferases are worth adding?
  • line 158 - it is not clear if alfa subunits of FTase and GGTase are encoded by the same gene or not.

To summarize, despite the small suggestions listed above, the article is highly valuable and gives an up-to-date and comprehensive review of the subject.

Reviewer 2 Report

This is a well-written article that provides comprehensive information on prenyltransferase enzymes, including their history, types, structure, cellular functions, and inhibitors. This review also broadly covers the detailed mechanisms of prenylation and other nuances, with an emphasis on prenyltransferases’ structural properties and interaction with substrate proteins. The illustration of catalytic mechanisms and secondary structures provides a great summary of text details and the mode of action of inhibitors discussed in section 6.

Suggestions

  • Given that the review stated the 4 broad classification of prenyltransferase inhibitors in line 619-622, authors should consider highlighting this classification in the list of inhibitors mentioned in Table 2-5.
  • Only brief mention is used of clinical trials. It would be helpful if this information was expanded. Also this information is almost entirely on binding and interaction, not activity or effect as well as reasons why the therapies haven't advanced to further stages of development.
  • Conclusions seem too focused on structure. I would like more on how this understanding of inhibitor structures could be useful for treating disease and how this information can be used for future study design.
